# Sex, strain, and lateral differences in brain cytoarchitecture across a large mouse population

David Elkind[1], Hannah Hochgerner[2], Etay Aloni[2], Noam Shental[1]*, Amit Zeisel[2]*

[1]Department of Computer Science, Open University of Israel, Raanana, Israel; [2]Faculty of Biotechnology and Food Engineering, Technion - Israel Institute of Technology, Haifa, Israel

*For correspondence:
shental@openu.ac.il (NS);
amit.zeisel@technion.ac.il (AZ)

**Competing interest:** The authors declare that no competing interests exist.

**Abstract** The mouse brain is by far the most intensively studied among mammalian brains, yet basic measures of its cytoarchitecture remain obscure. For example, quantifying cell numbers, and the interplay of sex, strain, and individual variability in cell density and volume is out of reach for many regions. The Allen Mouse Brain Connectivity project produces high-resolution full brain images of hundreds of brains. Although these were created for a different purpose, they reveal details of neuroanatomy and cytoarchitecture. Here, we used this population to systematically characterize cell density and volume for each anatomical unit in the mouse brain. We developed a DNN-based segmentation pipeline that uses the autofluorescence intensities of images to segment cell nuclei even within the densest regions, such as the dentate gyrus. We applied our pipeline to 507 brains of males and females from C57BL/6J and FVB.CD1 strains. Globally, we found that increased overall brain volume does not result in uniform expansion across all regions. Moreover, region-specific density changes are often negatively correlated with the volume of the region; therefore, cell count does not scale linearly with volume. Many regions, including layer 2/3 across several cortical areas, showed distinct lateral bias. We identified strain-specific or sex-specific differences. For example, males tended to have more cells in extended amygdala and hypothalamic regions (MEA, BST, BLA, BMA, and LPO, AHN) while females had more cells in the orbital cortex (ORB). Yet, inter-individual variability was always greater than the effect size of a single qualifier. We provide the results of this analysis as an accessible resource for the community.

## Editor's evaluation

The manuscript provides a new powerful tool as well as a large resource that should be useful both to the neuroscience community and more widely. The authors developed and applied a methodology to automatically estimate volume, cell number, and density of mice brains from multiple regions, by detecting the auto-fluorescence intensities of the cell nuclei. Using this platform, they analyzed a few hundred mouse brains available in the database of the Allen Mouse Brain Connectivity project. They identified strain-specific and sex-specific differences in several brain regions.

## Introduction

The mammalian brain can be divided into neuroanatomical units (i.e., brain regions) characterized by a shared function, connectivity, developmental origin, and/or cytoarchitecture (i.e., number and density of cells it contains). The mouse brain is the most extensively studied in mammals and its regions are well characterized. Although cytoarchitecture is one of the most prominent features of a brain region, few studies have systematically mapped cell bodies or quantified cell densities in

mouse brains, compared to the early, detailed cell mapping of the nematode *Caenorhabditis elegans* (*White et al., 1986*). On the other hand, extensive literature explored scaling of cell numbers and densities among mammalian brains (*Herculano-Houzel et al., 2006*; *Herculano-Houzel and Lent, 2005*; *Azevedo et al., 2009*). This however was performed by counting dissociated nuclei at the loss of deeper region-specific resolution. Obtaining an accurate cell count for a brain region is technically challenging. Previous estimates relied heavily on extrapolation from manual counting of 2D sections (stereology), making cell-resolved data for subcortical regions sparse (*Keller et al., 2018*). Analyzing complete brains using 2D histological sections remains labor-intensive because it requires sectioning, mounting, and accurate alignment with a reference atlas. Furthermore, automated cell counting proved particularly difficult in dense regions, such as the hippocampal formation (HPF) and the cerebellum (*Attili et al., 2019*). Automated block-face imaging methods solved several of these issues and drastically improved throughput (*Ueda et al., 2020*), For instance, serial two-photon tomography (STPT) (*Ragan et al., 2012*) was a technological breakthrough that integrated tissue sectioning with top-view light microscopy. STPT provided high-quality imaging in an optical plane below the sectioning surface and solved many problems of section distortion and atlas alignment, further easing downstream analysis. Yet, STPT typically represents a subsample of the complete volume, and some interpolation is needed.

Because of their limited throughput, histological studies cannot supply the number of analyzed brains needed to uncover potential variability between individuals, experimental conditions, and populations. Complementary approaches aimed at evaluating variability, for example, magnetic resonance imaging (MRI), can measure some features, such as the volume of given brain regions, and can even track individuals over time in a noninvasive manner. Yet, MRI lacks the accuracy needed for counting cells or assessing cell densities, and it remains difficult to simultaneously analyze regional volume *and* cell density with high accuracy brain-wide, especially in the large throughput required for comparing two experimental populations (such as two strains, or males vs. females). Therefore, there is a need for systematic measurement of all cells over hundreds of brains from multiple experimental groups.

To address this knowledge gap, we harnessed the Allen Mouse Brain Connectivity Project (AMBCA) dataset, which is largest existing cohort of whole-brain STPT images, produced by the Allen Institute for the different purpose of mapping mouse regional connectivity (*Oh et al., 2014*; http://help.brain-map.org/display/api/Allen%2BBrain%2BAtlas%2BAPI). We applied a deep neural network (DNN) to discern cell nuclei using the AMBCA background autofluorescence channel. This enabled us to perform a systematic brain-wide cell density estimation across hundreds of mouse brains. Based on the alignment with the Allen Mouse Brain Atlas (AMBA), we were able to simultaneously measure volume *and* density for each brain, for each region, over a large population. We constructed a comprehensive database that aggregates these results and provides them as an accessible resource to the community. We also discovered nontrivial relationships between densities and volumes, and gained insights into strain- and sex-dependent characteristics across various brain regions.

## Results

### Autofluorescence of STPT images displays cell nuclei

The AMBCA project was first published in 2014 (*Oh et al., 2014*). The project has systematically imaged 2992 full brains, using serial two-photon tomography, for the purpose of tracing neuronal projections and mapping regional (mesoscale) connectivity with GFP-labeled viral tracers. Each brain in the dataset is covered by 130–140 (median 137) serial coronal sections, with a gap of 100 μm, as reported in the AMBCA study (*Oh et al., 2014*). We noticed that the red (background) channel of STPT images, taken for the purpose of atlas alignment, typically features dark, round-like objects resembling cell nuclei. This phenomenon was described in previous literature (*Wang et al., 2020*; *Costantini et al., 2021*; *DaCosta et al., 2005*; *Kretschmer et al., 2016*). In particular, *Zipfel et al., 2003* characterized the use of multiphoton-excited native florescence and second harmonic generation for the purpose of staining-free tissue imaging. To confirm that these dark objects indeed represent cell nuclei with lower autofluorescence intensity than the surrounding lipid-rich brain tissue, we performed a standard 4% paraformaldehyde (PFA) perfusion-fixation followed by cryosectioning and nucleus (DAPI) counterstaining. In these sections, we observed the same low-autofluorescent objects

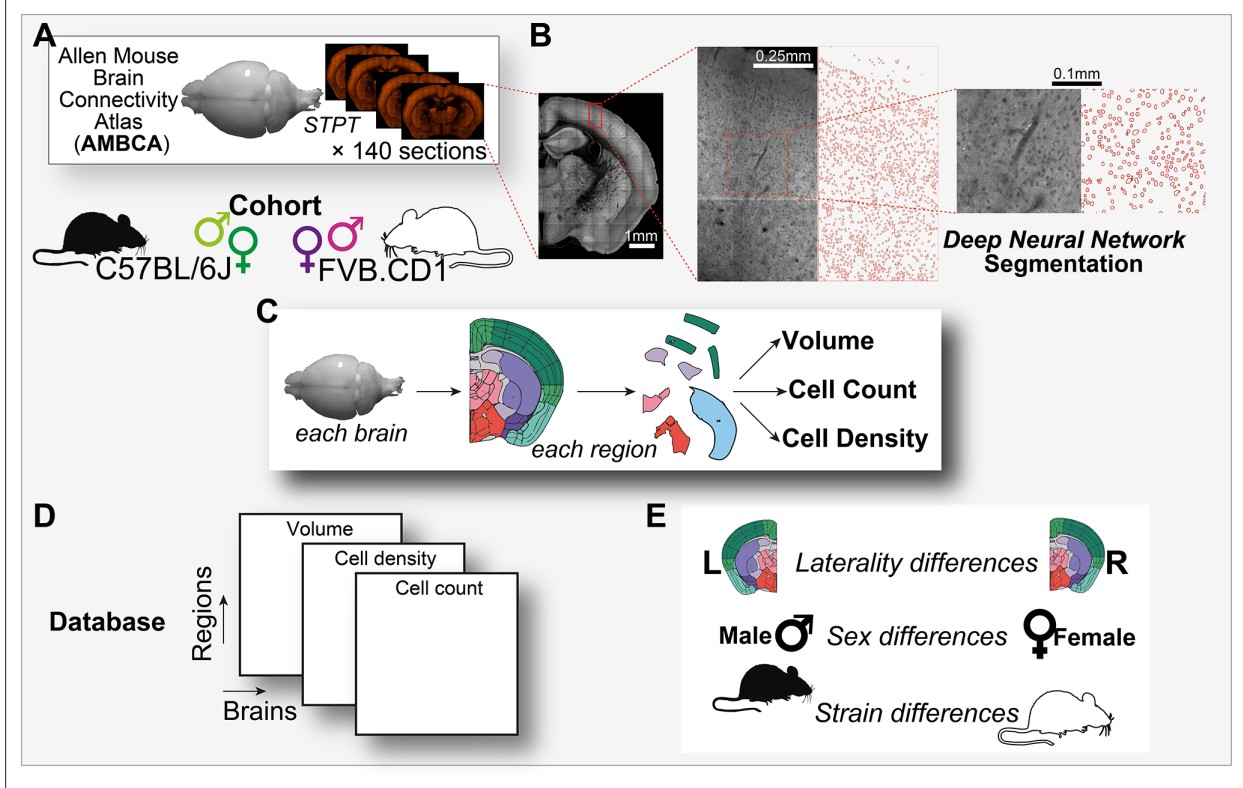

**Figure 1.** Graphical abstract. (**A**) Analysis is based on a cohort of 507 mouse brains from the Allen Mouse Brain Connectivity Atlas (AMBCA), males and females, of C57BL/6J and FVB.CD1 strains. Each brain was imaged in serial two-photon tomography (STPT) and comprises ~140 coronal sections spaced 100 μm apart along the anterior–posterior axis. (**B**) Example of nucleus segmentation in the isocortex. Each section was divided into tiles of 312 × 312 pixels (109 × 109 μm) (zoom-ins, right). A deep neural network cell segmentation model (see 'Methods') was applied to detect the contours of nuclei for downstream analysis across tiles, sections, and whole brains, as shown. (**C**) For each brain and region that passed QC (see 'Methods'), volume, cell density, and cell count were computed, resulting in a comprehensive database (**D**) available through our GUI. (**E**) The measured variables displayed region-specific laterality differences, sex and strain differences across the population.

The online version of this article includes the following figure supplement(s) for figure 1:

**Figure supplement 1.** Autofluorescence signal corresponds to nucleus validation.

**Figure supplement 2.** Discarding whole brains or regions of lower technical quality.

using epifluorescence microscopy. 91–98% of detected objects were also marked by DAPI, confirming that dark objects in STPT indeed represent cell nuclei (*Figure 1—figure supplement 1*). About 2–9% of detected low-autofluorescent objects were additional 'false positive' detections that were not obvious in the DAPI image. On the other hand, 26–45% of DAPI-detected nuclei were not observed in the autofluorescent images (false negatives), pointing to an underestimate of cell counts based on low-autofluorescence objects, compared to nuclear staining. A more systematic comparison between autofluorescence images and nuclear staining appears in the next section.

## Population-wide, regionally resolved exploration of neuroanatomical features

To automatically collect cytoarchitecture data for each brain, we trained a DNN model to detect and segment the nuclei (low-autofluorescent objects) in all brain regions, including those of the highest density, such as the dentate gyrus (DG). Because of computing constraints, we applied the model systematically to segment a subset of the AMBCA dataset comprising 507 brains (*Figure 1A and B* and 'Methods'). The model performed with an estimated 97% cell detection accuracy on a test set, with a false positive rate of <0.01 (see 'Methods') whenever image quality was sufficient (for exclusion criteria of whole brains or certain regions within sections, see 'Methods'). Using detected cells in each section, we obtained a local estimate of the volumetric cell density (see 'Methods'), which,

combined with the pixel-wise registration of brain regions provided by the AMBA, allowed us to estimate the average cell density per region for each brain. Similarly, we evaluated the per-region volume of each brain by linear interpolation across all sections (see 'Methods'). The anterior-most olfactory bulb (MOB) and posterior-most cerebellum (CB) were truncated in imaging, which likely led to an underestimate in their quantifications, and slightly higher variance compared to other regions of similar volume (*Figure 2—figure supplement 1D and E*). We further corrected a batch effect in the AMBCA dataset showing a small difference in overall brain volume across experimental batches (see 'Methods,' *Figure 2—figure supplement 1B and C*). In sum, we simultaneously estimated the 3D cell density ($D$) and volume ($V$) of each region for each brain (see 'Methods'). In total, we estimated per-region $D$ and $V$ for 532 basic regions annotated in the AMBA, which corresponds to levels 6–8 of the AMBA region hierarchy.

Cell count ($N$) is the product $V \times D$; therefore, it was not considered an independent variable. The median male C57BL/6J mouse brain contained a total of $76 \pm 11 \times 10^6$ cells, in 380 mm³ of gray matter, at a density of $2.05 \times 10^5$ cells/mm³. A pie chart of the volume and cell count of the main regions (level 4 of region hierarchy) calculated across 507 brains appear in *Figure 2B*, and absolute cell counts for C57BL/6J male mouse representative regions are shown in *Figure 2C*. We quantified each level of the hierarchical tree structure of the AMBA and found good correlation ($\rho = 0.86$ and $\rho = 0.98$ for log and linear scale, respectively; interclass correlation coefficient 0.98–0.99) with a recent 3D whole-brain single-cell resolved light-sheet microscopy study (*Murakami et al., 2018*; *Figure 2D*). The diameter of detected objects (nuclei) varied between 7 and 9.5 μm (*Figure 2E*, left), which at a nucleus/soma volumetric ratio of 0.08 (*Neumann and Nurse, 2007*; *Huber and Gerace, 2007*) corresponds to median cell body diameters from 16.25 μm in the RSPv6a, to 22 μm in the ENTl3. The regional variability of cell densities was high, ranging from $1 \times 10^5$ mm⁻³ in layer 1 isocortex (e.g., Mos1) to $6 \times 10^5$ mm⁻³ in the dentate gyrus granule layer (DG-sg). We show examples of regional distributions across the full cohort of 507 brains in the inset of *Figure 2E*, right.

The large number of AMBCA brains in our analysis enabled us to compare variabilities of macroscopic properties between subsets of the cohort, for example, to compare strains. We compared distributions of volume, cell density, and cell count at the coarsest hierarchical atlas level, that is, across gray matter cell groups in the brains of male C57BL/6J vs. male FVB.CD1 mice (*Figure 2F*). Median cell density was similar for the two strains, with considerably larger variance in FVB.CD1 males. FVB.CD1, however, had a 7.6% larger gray matter volume (GMV) than C57BL/6J. Combining these two features revealed an ~10% increase in the median cell count in FVB.CD1 vs. C57BL/6J (*Figure 2F*, right panel). These results suggest that (1) there is no simple relationship between volume and density, therefore, both properties should be measured simultaneously; and (2) a large cohort enables detection of relatively small differences.

To expand on our technical comparison between quantification based on autofluorescence vs. staining (*Figure 1—figure supplement 1*), we performed nuclear staining (Hoechst 33342) and whole-brain STPT imaging on nine brains of female C57BL/6J mice. We trained a DNN of the same topology using tiles from the resulting Hoechst images, registered with the AMBA coordinates, and repeated our per-region estimates of volume, density, and cell count (see 'Methods'). As expected, correlation between this in-house dataset and the AMBCA analysis was very high when comparing the volume across regions ($\rho = 0.99$, *Figure 2—figure supplement 2A*). In addition, cell count correlations were also high ($\rho = 0.99$, *Figure 2—figure supplement 2B*), but median cell count in Hoechst was 65% higher; recapitulating the trend toward false negatives observed in *Figure 1—figure supplement 1* (epifluorescence microscopy). However, agreement in cell density varied by region. The correlations in density were fair across cortical regions of layers 2/3, 4, 5, and 6a ($\rho = 0.78$, *Figure 2—figure supplement 2C1*): they displayed comparable densities between the methods (*Figure 2—figure supplement 2D1*). The correlation value across cortical regions 1 and 6b was similar ($\rho = 0.79$), yet the error in absolute values was significantly higher (*Figure 2—figure supplement 2C2*), at about twofold higher density using Hoechst (*Figure 2—figure supplement 2D1*). The correlation across brain stem regions was also lower ($\rho = 0.56$, *Figure 2—figure supplement 2C3*), yet rank order across regions was similar to cortical regions 1 and 6b. Cerebellar regions, which are significantly denser, displayed a twofold density difference ($\rho = 0.65$, *Figure 2—figure supplement 2C4*). This detection bias may therefore stem from several underlying causes; (1) inaccurate registration of border regions (CTX L1 and 6b); (2) physical detection limits of the low-autofluorescent objects compared to stained objects;

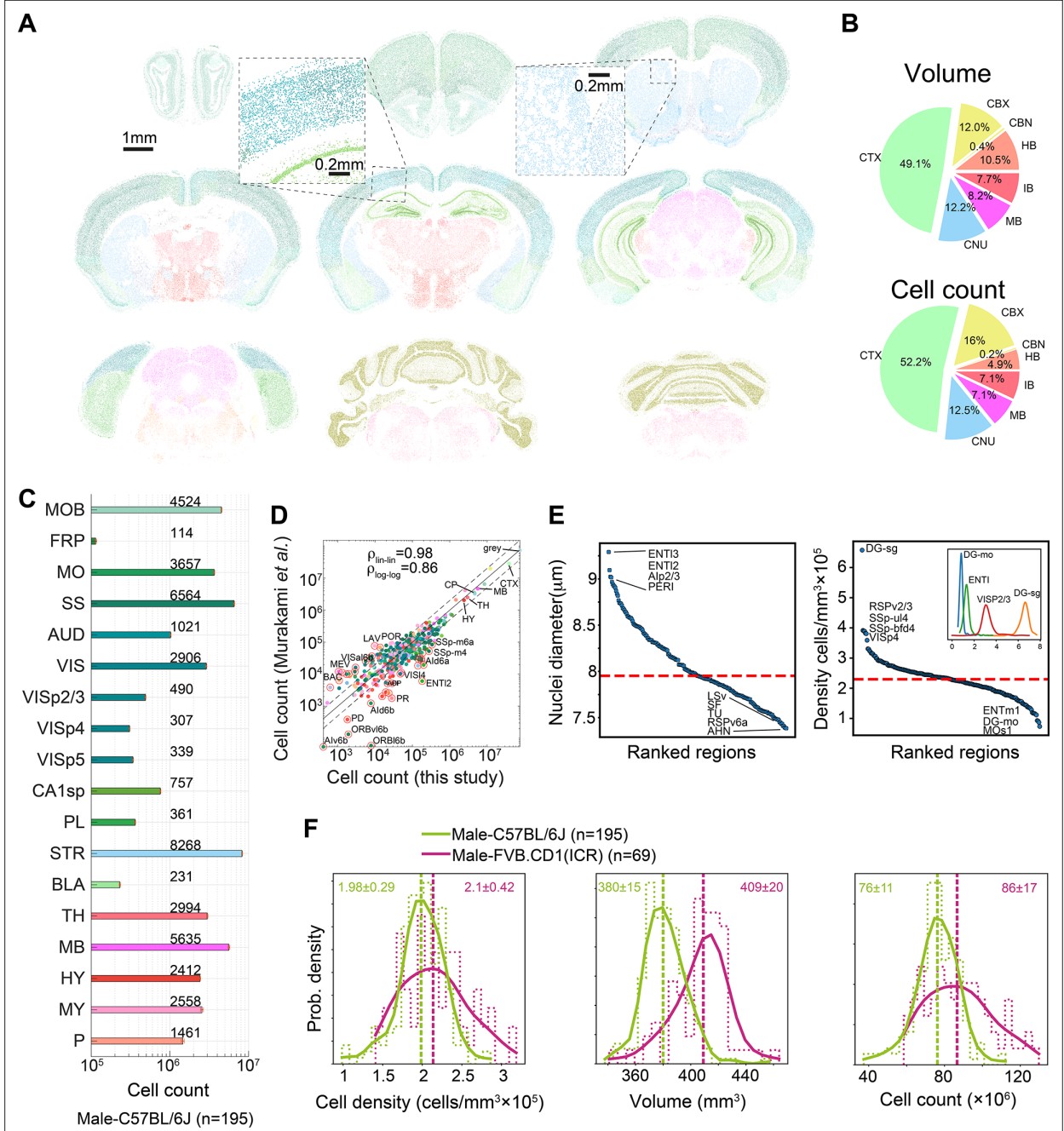

**Figure 2.** Survey of neuroanatomic properties of the mouse brain. (**A**) Segmentation of several sections of one particular brain; segmented nuclei are colored using the Allen Mouse Brain Atlas (AMBA) region convention. (**B**) Pie charts of the median volumes and cell counts across all 507 brains in the main brain regions, colored using AMBA nomenclature. (CTX: cerebral cortex; CNU: cerebral nuclei; MB: midbrain; IB: interbrain; HB: hindbrain; CBN: cerebellar nuclei; CBX: cerebellar cortex). (**C**) Median cell counts for selected brain regions in C57BL/6J males (number near bars in thousands; SEM is displayed per region yet values are very small). (**D**) Comparison of region cell counts between this study and Murakami et al., over C57BL/6J males; dots above/below the dashed lines represent regions with greater than twofold difference. The correlation coefficient in both linear and log scales is displayed. The intraclass correlation coefficient (ICC) values were 0.98–0.99 for six ICC forms. (**E**) Ranking of 532 regions by nucleus diameter (left) and density (right). Each dot corresponds to the median value of one region over 507 brains. Red dashed line, median across regions. Inset shows distributions of density over 507 brains for selected regions. (**F**) Distribution of cell density (left), brain volume (middle), and cell count (right), comparing C57BL/6J males and FVB.CD1 males across basic cell groups and regions ('gray'). Step-like dashed lines represent histograms while full lines correspond to kernel estimations of the probability density function. Dispersion values correspond to standard deviations. Source data for panels (**D, E**) is provided in *Figure 2—source data 1*.

The online version of this article includes the following source data and figure supplement(s) for figure 2:

*Figure 2 continued on next page*

*Figure 2 continued*

**Source data 1.** Comparison of cell count per region (male C57BL/6J) with *Murakami et al., 2018*; median cell diameter and cell density per region over all 507 brains in this study.

**Figure supplement 1.** Technical aspects of 3D cell counting.

**Figure supplement 2.** Comparison with quantification based on nuclear staining.

**Figure supplement 3.** Density and nucleus diameter along cortical regions.

and (3) region or cell type-inherent differences in autofluorescence, resulting in lower detection of cells in glial-rich (e.g., CTX L1, brainstem), compared to neuron-rich (e.g., other cortex layers) regions.

To test the power of our model, we explored the densities and nucleus diameter of cortical regions (*Figure 2—figure supplement 3*). First, we considered the HPF because imaging-based quantification of its denser regions (pyramidal layers of Ammon's horn and the granule layer of the dentate gyrus) has been difficult (*Attili et al., 2019*) and was achieved only recently (*Murakami et al., 2018*; *Seiriki et al., 2017*). Analyzing 195 C57BL/6J male brains, we found that the pyramidal layer of CA1 was denser than that of CA3 and CA2, whereas nucleus size was larger in CA3. In the dentate gyrus, the granule layer had the highest density of all regions, with >$6.5 \times 10^5$ cells/mm$^3$, and nuclei were largest in the polymorph layer (*Figure 2—figure supplement 3*, upper panels). In the isocortex, we examined the extent to which the cortical layers across cortical divisions differed in density and size (*Figure 2—figure supplement 3*, lower panels). Layer 1 was consistently underpopulated, having a density of about $10^5$ cells/mm$^3$. The overall rank order from densest to sparsest was maintained, with layer 4 > layer 6a > layer 5 > layer 2/3 > layer 1, suggesting a similarity in cytoarchitecture between cortical regions. Layer 4 of the primary visual and somatosensory cortices had higher density than did the auditory and visceral cortices. Nucleus diameters showed less distinct distributions between layers, although layer 2/3 and layer 5 tended to have larger nuclei than did layers 4 and 6a.

## Density differences between left and right hemispheres

Brain laterality has been discussed in the literature since Broca and Wernike found language dominance on the left side of the brain. Although evidence for brain laterality in mice is more scarce, *Levy et al., 2019* have suggested functional and circuit differences in the auditory cortex. We performed a systematic comparison of the left and right hemispheres seeking differences in region-wise volume and/or cell density. In C57BL/6J (n = 369), we found 229 regions with lateral cell density bias of over 5%, and up to 30%. Both sides showed similar numbers of biased regions in density (left, 100; right, 129, *Figure 3A*). Neighboring regions in prefrontal cortex ACAv2/3, ACAd2/3, ORBl2/3, and PL/23 were up to 30% denser in the right hemisphere. In addition, a more posterior part of the HPF, the parasubiculum (PAR) also showed a consistent per-brain higher cell density in the right hemisphere (*Figure 3B*). In contrast, cortical areas GU2/3, RSPv6a, VISC2/3, and AUDv2/3 showed more than 20% higher density in the left hemisphere. Strikingly, we found consistent density bias in left hemisphere cortical regions, specifically in layers 2/3 (*Figure 3A*, upper inset). This bias was both most consistent and pronounced in a group of neighboring ventrolateral areas: visceral, gustatory, temporal association, and auditory (*Figure 3C*), where the latter may be consistent with the Levy et al. findings. For example, in *Figure 3D* we show higher density in the right hemisphere visceral cortex layer 2/3, while corresponding layers 4, 5, and 6a were not biased. Laterally biased regions were consistent across strains (e.g., C57BL/6J vs. FVB.CD1, *Figure 3—figure supplement 1A and B*).

We note that volume laterality was less pronounced with some regions slightly larger in the right hemisphere (*Figure 3—figure supplement 1C*). Most regions did not show laterality in both density and volume (e.g., PAR and VISC2/3 in *Figure 3B and D* display density laterality, while no bias in volume is observed).

## Regions with volume/density sexual dimorphism in C57BL/6J mice

To examine whether differences in overall brain volume or density (*Figure 2F*) are isotropic, we conducted a region-specific analysis of volume, density, and cell count. Differences between males and females in regional neuroanatomy have been extensively described, including dimorphic volume and cell count in the medial amygdala (MEA) (*Morris et al., 2008*; *Morris et al., 2005*) and in the bed nuclei of the stria terminalis (BST) (*Garcia-Falgueras et al., 2005*). We first compared C57BL/6J

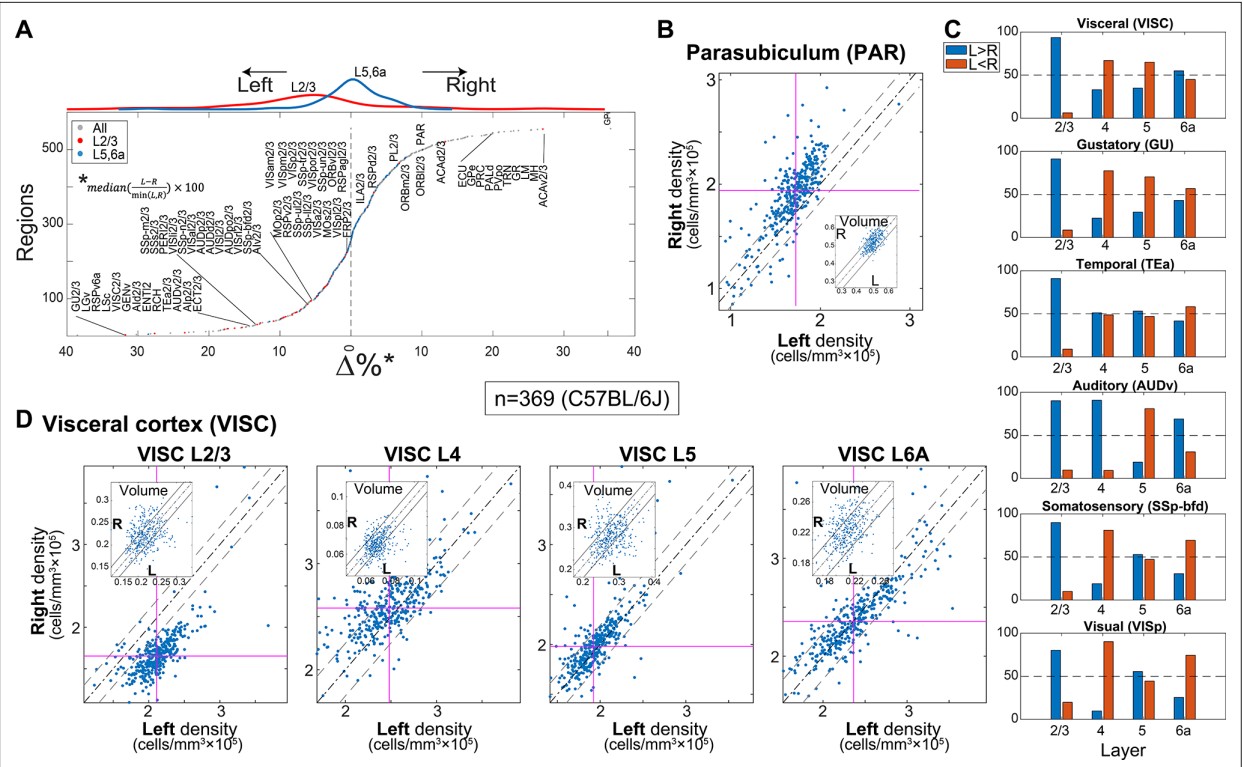

**Figure 3.** Region-wise laterality of cell densities in C57BL/6J. (**A**) Laterality differences in density (all C57BL/6J brains n = 369) shown for all regions whose volume >0.05 mm³ (excluding layers 1 and 6b) resulting in 559 regions. Regions are sorted by their bias to the left. Red and blue dots show layer 2/3 and layer 5/6a, respectively. Upper inset shows the distribution of layer 2/3 regions and layer 5/6a in red and blue lines, respectively. (**B**) Scatter plot of the density of the left vs. right hemisphere for parasubiculum (PAR). Each dot corresponds to one brain. Vertical and horizontal cyan lines correspond to the left and right median values, respectively. The dashed dotted line is equi-density value and the dashed line corresponds to a 10% offset by the median of the left density. The inset shows an analogous scatter plot for volume, with no observed lateral bias. (**C**) Percentage of brains with tendency for left or right hemisphere density across cortical areas (blue, left > right; orange, right > left). (**D**) Scatter plots of the density of the left vs. right hemisphere for visceral area layers 2/3, 4, 5, and 6A.

The online version of this article includes the following figure supplement(s) for figure 3:

**Figure supplement 1.** Volcano plots showing region-wise comparison of left vs. right hemispheres in C57BL/6J and FVB.CD1.

males (n = 152) with females (n = 140). Although at the global level (whole-brain gray matter), males and females had similar volume (380 ± 17 mm³ and 380 ± 14 mm³, for females and males, respectively), in density (1.97 ± 0.28 × 10⁵ cells/mm³ and 1.97 ± 0.28 × 10⁵ cells/mm³ for females and males, respectively) and total numbers of cells (75 ± 10 × 10⁶ and 77 ± 9 × 10⁶ for females and males, respectively) (***Figure 4A***), certain regions displayed sexual dimorphism in one or more property. We conducted rank-sum testing on each region that passed QC (see 'Methods') for sex differences, in volume, density, or count (***Figure 4B***). Most regions were consistent with the overall trend of equal volume, yet MEA and BST, but also the ventral cochlear nucleus (VCO), taenia tecta ventral (TTv), and the medial preoptic nucleus (MPN) were >5% larger in males. A handful of regions, for example, ventrolateral orbital area layer 2/3 (ORBvl2/3), displayed the opposite effect and had larger volume in females. ORBvl2/3 was the only region to also display significantly higher density and cell count in females. In contrast, many regions were 5–20% denser in males, which also resulted in higher cell counts. These regions include other parts of the amygdala (BMA, BLA, and the CEA), hypothalamic regions LPO and AHN, and cortical regions such as the primary auditory layer 2/3 (AUDv2/3). ***Figure 4—source data 1*** specifies whether each region is significantly dimorphic in volume, density, or count.

Next, we looked beyond the rank-sum statistical test, governed by the median of the distribution, at examples of how distributions differ in volume and/or density. For example, ORBvl2/3 showed both larger volumes and slightly higher density in females (***Figure 4C***, left), resulting in significantly more cells in females (***Figure 4A***). The opposite was the case for BST, where males had larger volume

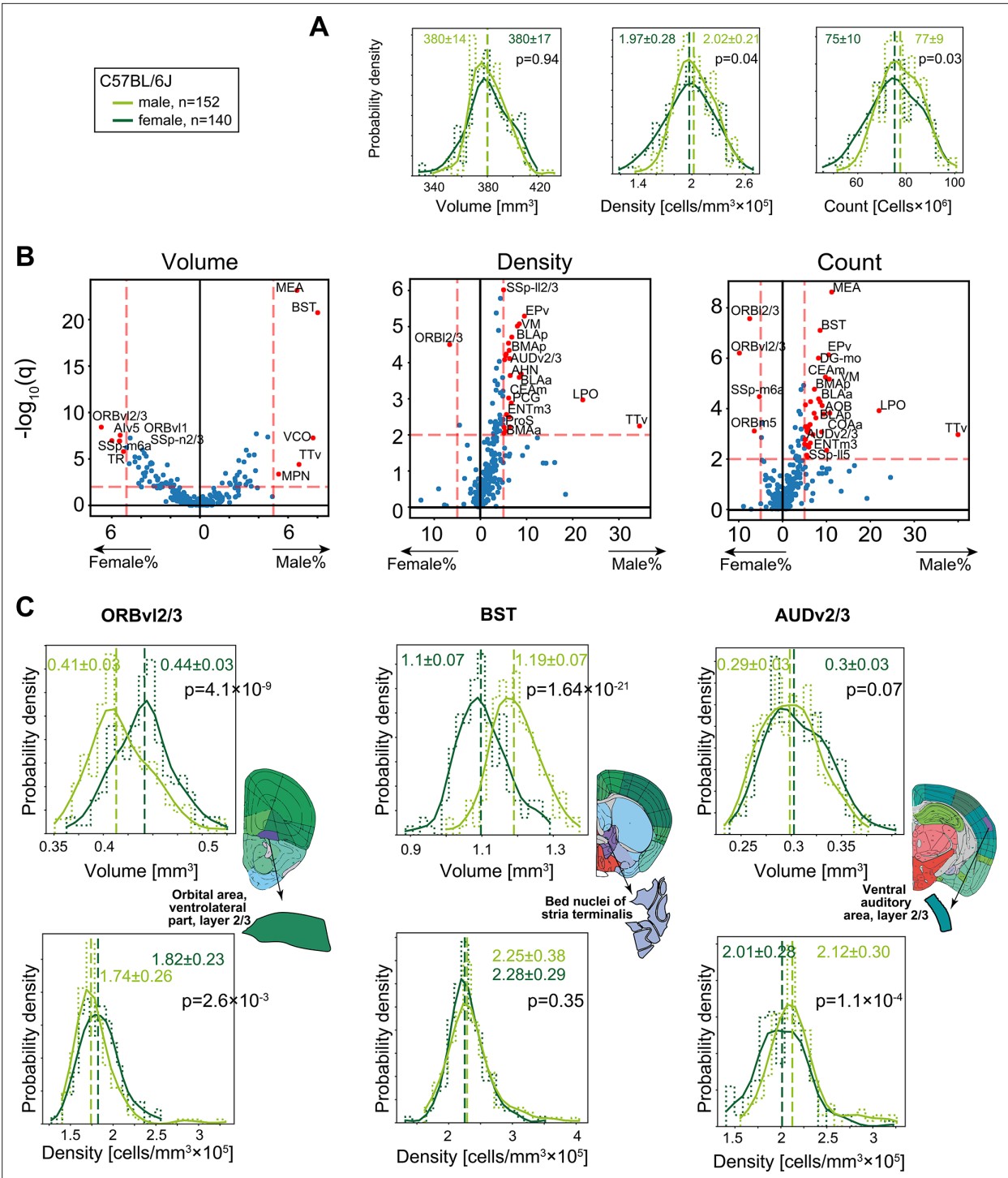

**Figure 4.** Sexual dimorphism in C57BL/6J. (**A**) Distribution of volume (left), density (middle), and cell count (right) for the whole brain gray matter ('gray') in female (dark green), and male (light green). p-Values correspond to a rank-sum test. Step-like dashed lines represent histograms while full lines correspond to kernel estimations of the probability density function. Dispersion values correspond to standard deviations. (**B**) Volcano plots showing per-region statistical testing for male versus female difference in volume (left), density (middle), and cell count (right), each dot representing one region. Horizontal axis, median differences (%); vertical axis, q-values (FDR-corrected rank-sum p-values by BH procedure in -log10 scale). Red dots highlight regions with an effect size larger than 5% and q < 0.01. Source data for this panel is provided in *Figure 4—source data 1*. (**C**) Examples of regions that display sexually dimorphic volume and/or density. Distributions of volumes appear in the upper row, distributions of densities in the lower row.

The online version of this article includes the following source data and figure supplement(s) for figure 4:

*Figure 4 continued on next page*

*Figure 4 continued*

**Source data 1.** Median values per region for cell count, volume and cell density for male and female C57BL/6J or FVB.CD1.

**Figure supplement 1.** Validation and additional aspects of sexual dimorphism.

yet similar density (*Figure 4C*, middle). As a third example, we showed the case of AUDv2/3, which displayed no difference in region volume, yet density in the male brains was higher (*Figure 4C*, right). In an independent AMBCA cohort of additional 663 males and 166 females, we could robustly replicate our findings regarding region-specific sexual dimorphism: volumes of MEA, BST, and ORBvl2/3 displayed sex differences of consistent effect size and variance (*Figure 4—figure supplement 1A–E*).

In sum, the population-wide survey revealed a number of sexually dimorphic areas; Yet, to what extent were region volumes or densities predictors of the individual's sex? To answer this question, we trained linear support vector machine (SVM) classifiers. In brief, we randomly selected 2/3 of the C57BL/6J brains (n = 184), where each brain was represented by a 532-dimensional vector of either volume or density across regions, and trained the SVM. We then tested its performance on the remaining 1/3 of brains (n = 97). This process was repeated 100 times and resulted in an average accuracy of 78 and 90% for density and volume, respectively. From this, we identified the regions that had the highest contribution to the separating hyperplane (*Figure 4—figure supplement 1F and H*) and trained new classifiers based on these top regions, adding one region at a time. Using volume data, MEA and BST alone performed classification to >78% accuracy. Adding the postpiriform transitional area (TR) and the rostral lateral septal nucleus (LSr), the classifier's performance saturated to about 90% (*Figure 4—figure supplement 1G*). *Figure 4—figure supplement 1J* shows the SVM separating line based on four pairs of regions (e.g., BST and LSr) yielding an accuracy of 79–86%. In contrast, the density-based classifier showed only incremental improvements when adding almost any of the top 10 regions (*Figure 4—figure supplement 1I*). Together, these results suggest that although many brain regions show significant differences between males and females both in volume and density (*Figure 4*), the considerable overlap between the male and female population distributions in each of these regions hampers classification. Apart from MEA and BST, the classifier revealed TR, LSr, MPN, and ORBl2/3 as predictors for sex based on volume, while density-based classification was based on amygdala-related regions (COApl, BMAp) and frontal cortical regions (ILA and ORB).

## Strain differences in volume and density

We next investigated the relation between recorded body weight and GMV. To this end, we added the cohort of outbred FVB.CD1 mice, a strain with 40–50% higher body weight than C57BL/6J. As expected, in both strains, males and females showed distinct distributions for body weight, and males were larger than females (*Figure 5A*). Within each strain, body weight did not correlate with gray volume. We next quantified sex and strain differences in brain volume and density, resolved to neuroanatomical regions. First, we compared strain differences in females with those in males, showing concordance/discordance patterns between males and females (sex) (*Figure 5B and C* and schematic to the right). Second, we compared sex differences in FVB.CD1 with those in C57BL/6J, showing concordance/discordance patterns between strains (*Figure 5D and E* and schematic to the right).

### Strain-wise analysis

FVB.CD1 brains were overall larger, but the volume expansion with respect to C57BL/6J was not uniform across regions. Region volumes ranged up to 30% differences, with the extreme example of the cerebellum (CENT2), whose size increased by 45% in both FVB.CD1 males and females (*Figure 5B*). Moreover, per-region volume differences between strains were, in general, larger in males (i.e., most data points in *Figure 5B* quadrant III are above the diagonal). Only two regions showed larger volumes in C57BL/6J: the main olfactory bulb (MOB) and the lateral amygdalar nucleus (LA) (*Figure 5B*, quadrant I).

A similar comparison for cell density per region suggests nonuniform density differences, with most regions being denser in FVB.CD1 (*Figure 5B*, quadrant III). Notably, olfaction-related regions (AOB and MOB) showed higher density in C57BL/6J.

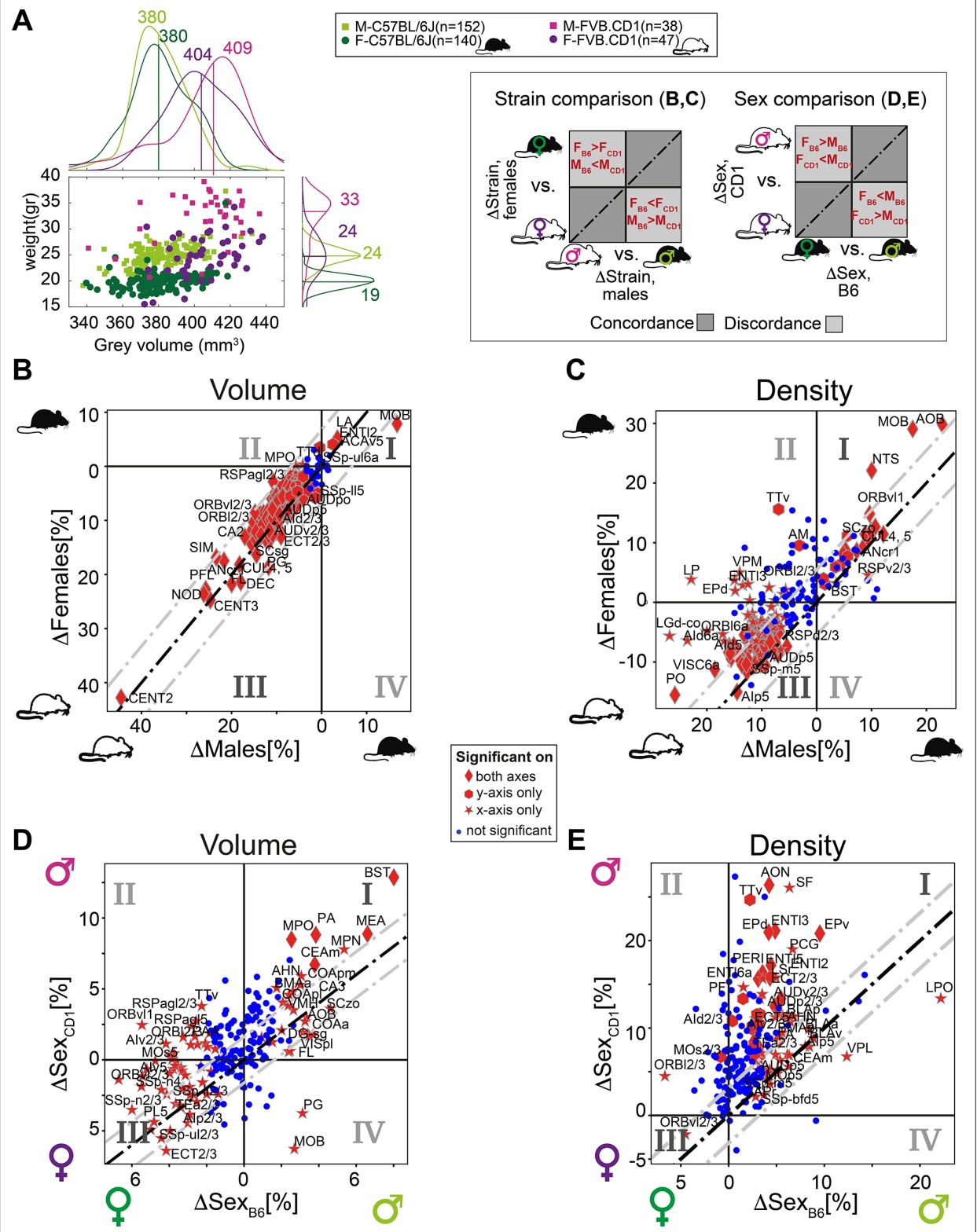

**Figure 5.** Sexual and cross-strain dimorphism in C57BL/6J (B6) and FVB.CD1 (CD1). (**A**) Scatter plot showing body weight vs. gray volume for 507 brains. Side panels show the group distributions of gray matter volume (upper) and weight (right). Lines are the medians whose values are indicated. (**B, C**) Strain comparison of per-region volume (**B**) and density (**C**). Differences between the median values of the strains, per region, are shown for males (horizontal axis) and females (vertical axis). Points in quadrants I and III suggest concordance between males and females across strains, as illustrated in the schematic on the left. Red markers designate statistical significance (q-value < 0.05) in either axes or in both. (**D, E**) Sex comparison of per-region

*Figure 5 continued on next page*

*Figure 5 continued*

volume (**D**) and density (**E**). Points in quadrants I and III suggest concordance between C57BL/6J and FVB.CD1 across sex, as illustrated in the schematic on the left. Source data for panels (**A–E**) is provided in *Figure 5—source data 1*.

The online version of this article includes the following source data for figure 5:

**Source data 1.** Volume and body weight per mouse; strain differences in volume or cell density for male and female; sex differences in region volume or cell density for C57BL6 and FVB.CD1.

## Sex-wise analysis

Differences in volume confirmed sexual dimorphism in MEA and BST, which were larger in males for both strains. These differences were more pronounced in FVB.CD1 than in C57BL/6J (*Figure 5D*, quadrant I). Many brain regions showed 'strain-discordant' dimorphism, with females having a larger volume in C57BL/6J and males in FVB.CD1 (*Figure 5D*, quadrant II). Although total brain volume in FVB.CD1 males was larger, some regions showed larger volume in females (e.g., the previously mentioned orbital cortex ORB, *Figure 5D*, quadrant III). Comparing sexual dimorphism in density (*Figure 5E*), we found a simpler and more consistent picture: in both strains, males had higher density in most regions (except for, e.g., ORBvl2/3). Note that in density as well, sex differences were found to be larger in FVB.CD1 (most data points in *Figure 5E*, quadrant I, are above the diagonal).

## Region-wise correlations between volume and density across brains

To the best of our knowledge, no previous study simultaneously quantified cell density (*D*) and brain region volume (*V*). We therefore sought to investigate whether constraints exist between *D* and *V*. For example, if the number of cells in a region is constant across brains, *D* and *V* must be negatively correlated. If, by contrast, the number of cells in a region, *N*, scales with the volume while *D* remains constant, *D* and *V* display zero correlation. If a positive correlation exists between *D* and *V*, the number of cells *N* grows faster than linear with respect to either *D* or *V* (*Figure 6A*). Based on per-region measurements of both *V* and *D*, we calculated regionally resolved Pearson correlations between volume and density in C57BL/6J (*Figure 6B*). In 79% of regions (356/451), cell density was negatively correlated with volume (*Figure 6C*), with a median correlation of –0.14. For example, we showed two regions where *N* was positively correlated with both *D* and *V*, yet the correlation between *D* and *V* was either positive (AAA, *Figure 6D*) or negative (SSs2/3, *Figure 6E*). This suggests that across regions cell count does not scale uniformly with volume.

## Inter-brain similarity between regions based on volume and density

Finally, we assessed similarity between regions, based on volume or density. We used tSNE as a 2D embedding method over the density data (*Figure 6F and G*; the similarity matrix that served as input to tSNE appears in *Figure 6—figure supplement 1C*). Briefly, each region is characterized by a vector of 204 components, each representing its density across one brain. 2D embedding aims to preserve the local similarity between regions. The density-based tSNE embedding map in *Figure 6F* reveals clear 2D 'clusters,' largely consistent with neuroanatomical classification. Cortical regions appear in the upper part of the map (colored green), and cerebellum (yellow), midbrain, and hindbrain in the lower part. We further explored whether the order within the cortical part may be explained by layer structure or by cortical division, but found no clear structure (*Figure 6—figure supplement 1A and B*). Compared to *Figure 6F*, tSNE embedding based on volume was more 'dispersed' and displayed disorder with respect to neuroanatomical classification (*Figure 6G*). To demonstrate that the tSNE map is indeed based on true variations in region-to-region correlations, we compared density-based and volume-based correlations. First, for each region we identified its 10 most correlated regions based on either density or volume. These correlation values were higher for density-based correlations across almost all regions (*Figure 6—figure supplement 1D*), supporting the observed density-based 'order.' Second, we selected 10 representative regions across the brain and calculated all their pairwise correlations (*Figure 6—figure supplement 1E*), showing that even for distant regions, density-based correlations remain much higher than volume-based correlations. Thus, similarity in volume across regions is less 'preserved' brain-wide than similarity in cell density.

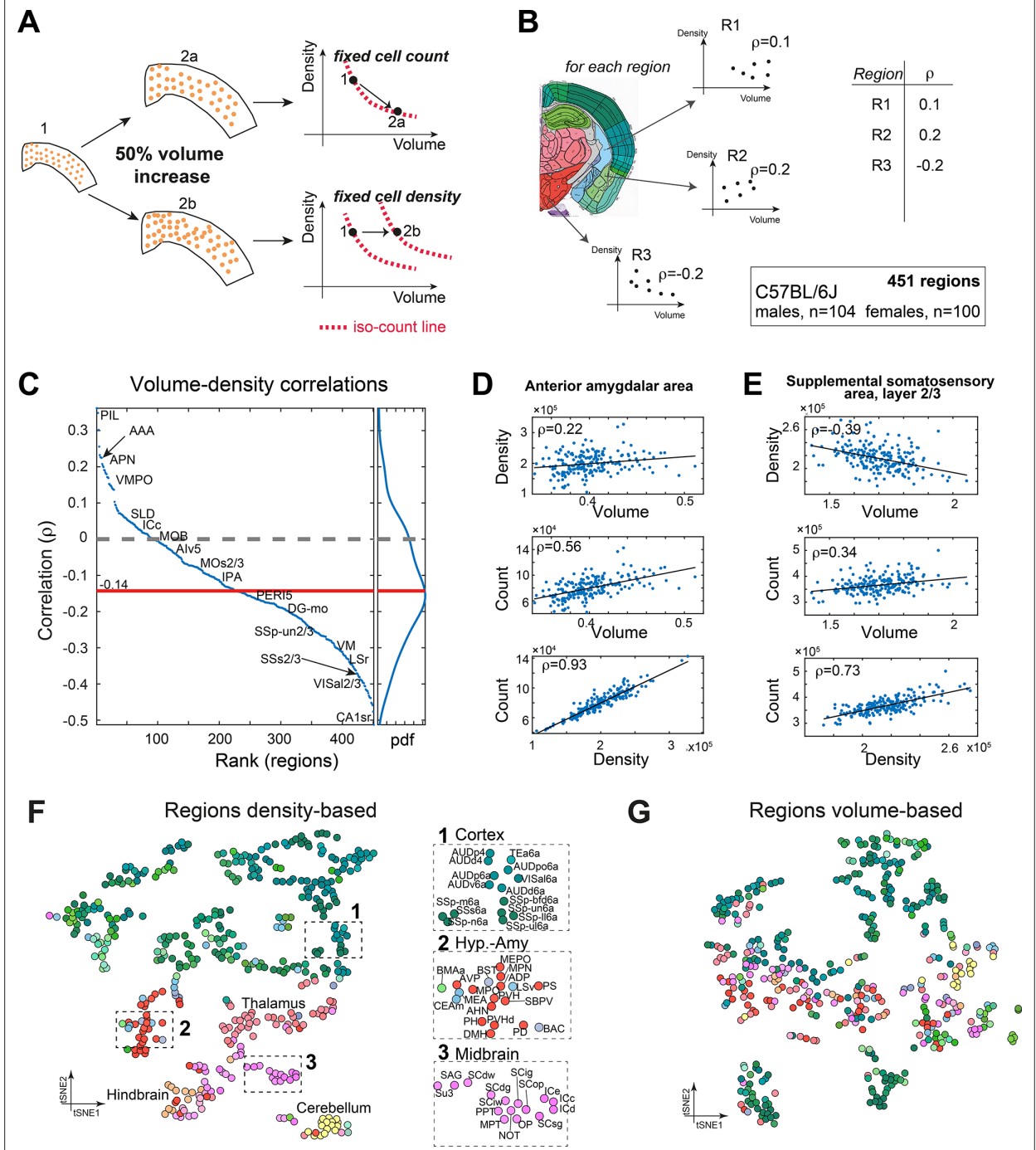

**Figure 6.** Correlations between volume, cell count, and density. (**A**) Schematic illustration of two types of relations between regional cell density and volume, associating region expansion with a fixed number of cells (upper) or with a fixed density (lower). Each regional expansion can be represented by a shift in the volume–density plane (right column). (**B**) A scheme showing how for each region the correlation between density and volume was measured over the whole dataset. (**C**) Brain regions ranked by the correlation between volume and density based on C57BL/6J (104 males, 100 females). Correlations higher than 0.14 or lower than –0.14 correspond to q-values lower than 0.05. Side panel displays the distribution of correlation values, and its median is denoted by the red line. Arrows point to regions mentioned in the following panels. (**D**) Correlations between volume, density, and count in the anterior amygdalar area (AAA). (**E**) As (**D**), for supplemental somatosensory area, layer 2/3 (SSs2/3). (**F, G**) Visualizing similarity between brain regions across the population, by tSNE embedding on pairwise correlations between region density (**F**) or region volume (**G**). Each dot represents a region and is colored according to the Allen Mouse Brain Atlas (AMBA) convention. Dashed rectangles indicate zoom-ins on three frames on each tSNE.

The online version of this article includes the following figure supplement(s) for figure 6:

**Figure supplement 1.** Region to region correlation within the C57BL/6J population.

# Discussion

We presented an automated, imaging-based, staining-free study of neuroanatomy and cytoarchitecture in the mouse brain. We conducted our measurements on a massive, high-quality dataset of serial two-photon tomography (*Oh et al., 2014*), aligned with a well-annotated reference atlas (*Wang et al., 2020*). This made possible, for the first time, a detailed population-wide analysis of two important neuroanatomical variables simultaneously: cell density and volume, resolved for 532 regions. The data spans an unprecedented cohort of 507 mice of two strains, the inbred C57BL/6J, and the hybrid FVB1.CD-1, each represented by both females and males. Our high-throughput measurements of cell densities were achieved by using a DNN trained to detect low-autofluorescent cell nuclei with high accuracy, even in the most cell-dense regions of the brain.

The study has several limitations. First, the model is sensitive to the contrast between dark nuclei and autofluorescent surroundings, which can be limited by image quality and tissue composition. In particular, the staining-free approach depends on the presence of intrinsic molecular indicators such as NADH, retinol, or collagen (*Zipfel et al., 2003*), which may vary between cell or tissue components, even within the brain. In the hindbrain (pons, medulla), contrast was exceedingly weak, and we expect our quantifications in this region to be 66% of the value estimated by nuclear staining (*Figure 2—figure supplement 2*). Second, AMBA annotations were not always resolved to the most refined level of the atlas hierarchy. For example, density values for the cerebellum appear to be uncharacteristic because the cell-dense granule layer and sparse molecular layer were not distinguished at the deepest level of annotation (e.g., CENT3 included the granular and molecular layers). The same is true for the hippocampus CA1-2-3, where we used cell density-based clustering to distinguish the pyramidal layer (sp) from its surrounding sparse layers (slm, so, sr, see 'Methods'). Therefore, although the model performed exceedingly well even in these cell-dense regions, the absence of annotations stood occasionally in the way of making biologically meaningful distinctions. Third, regions at the anterior–posterior edges, that is, olfactory bulb (MOB) and cerebellum (CB), appear partially truncated, and we likely report an underestimate of their parameters. MOB and CB volumes also displayed slightly higher variance than more interior regions with similar volume. Therefore, their population comparisons would appear noisier (although likely not biased), compared to other regions.

We further attempted to estimate the region-specific accuracy of our cell counting by comparing autofluorescence STPT with brain-wide imaging of nuclear-stained STPT. However, this comparison is technically nontrivial because of the native physical properties of direct staining vs. autofluorescence. For example, stained nuclei located off the focal plane may appear in the image, yet remain undetected by autofluorescence. In addition, tissue composition (e.g., cell types, extracellular matrix) may affect the imaged region. Indeed, in regions rich with non-neuronal cells the error of autofluorescent-based counting was larger compared to nuclear staining. Hence, one may speculate that autofluorescent-based detection is biased for neurons.

Nevertheless, we provided key statistics that help answer fundamental, recurring questions in neuroanatomy. Although no other study presented simultaneous measurements of volume and cell density, our data correlate well with a wealth of literature in the field. We achieved good region-wise correlation with full 3D volumetric cell counts by expansion microscopy (*Murakami et al., 2018*; *Figure 2D*). Our derived cell count of mouse brain gray matter ($76 \pm 11 \times 10^6$ for male C57BL/6J) is well within the range of existing cell count estimates for adult males ($67 - 150 \times 10^6$ cells) (*Murakami et al., 2018*; *Seiriki et al., 2017*; *Herculano-Houzel et al., 2011*; *Herculano-Houzel et al., 2015*). And although the current mouse brain dataset is unique in its scale, we foresee the application of full-brain, staining-free STPT and DNN-led neuroanatomical characterization could benefit the study of disease models, or be applied to other species for comparative neurobiology, as has been done in dissociation-based studies that revealed mammalian brain scaling rules (*Herculano-Houzel et al., 2006*; *Herculano-Houzel and Lent, 2005*; *Azevedo et al., 2009*).

By measuring the largest cohort to date, we provided partial support for the notion that this extreme range in the literature may not stem from variation in strain or sex, but rather from individual differences (*Murakami et al., 2018*). The median cell counts between sexes and strains differed no more than 13% overall, or ~40% for the most deviant individual structures (MOB and CENT2), while the standard deviation across individuals was ~10 million cells (for C57BL/6J male alone). Hence, values between $55 \times 10^6 - 95 \times 10^6$ are within ±2 SDs of the distribution of total grey matter cell

counts. We claim that the notion of 'ground truth' values of brain cell number *can* be reached, yet are best reflected by a population distribution.

Our dataset provides a large, important corpus toward this 'ground truth,' and similar studies can further help distinguish technical biases from biological variation. For example, our cohort provides the first brain-wide survey of cell architecture laterality, with numerous regions showing density bias to the right or left hemisphere; and some volume bias in favor of the right hemisphere. We describe a remarkable phenomenon of ventrolateral cortical areas increasing cell density in left hemisphere layer 2/3, but no other layers, that may be interesting to follow up functionally.

We further validated and expanded on the existing literature describing examples of regions that show sex- or strain-based differences. Sexual dimorphism of brain regions was significant, as exemplified by the success of our SVM-classifier, yet the ability to separate males from females was limited due to population's interindividual differences. For instance, medial amygdala and bed nuclei of the stria terminalis were both larger and denser in males, but to a lesser extent than reported in smaller studies (*Cooke et al., 2007*) and to a similar extent to what was reported in MRI-based studies (*Qiu et al., 2018*). By contrast, in females, several prefrontal cortex structures were larger (e.g., ORBvl2/3), which resulted in higher cell counts. We found no evidence of this phenomenon in the literature on mice, but an MRI population study of 2838 human individuals found higher GMV in prefrontal areas in women (*Lotze et al., 2019*). Between the strains, we found considerable differences in the olfactory system, which was larger and denser in C57BL/6J, and in the cerebellum, which was larger in FVB. CD-1. Finally, we provide an accessible, web-based platform for open exploration of the data. The web application allows researchers to freely compute distributions of any measured neuroanatomical features, for any brain region, and across the entire population or specific subsets. This exploratory resource can be of great use for experimental design and lead to more accurate brain modeling.

## Methods
### Data
The AMBCA dataset (*Oh et al., 2014*) consists of 2992 brains, of which we processed 507 and eventually used 378 in our analysis (the strain and sex breakdown of the brains are shown in *Table 1*). Each brain consisted of ~140 section images captured every $100\mu m$ along the anterior–posterior axis using two-photon tomography (*Ragan et al., 2012*). Image resolution was $0.35\mu m$ per pixel. AMBCA post-processed section images for noise removal. Rather than using the red, green, and blue channels occupied by the fluorescent reporter for anterograde connectivity tracing, we used the background channel of the images, as provided by AMBCA, without additional processing, except for converting the RGB image to grayscale.

### Training a deep neural network for cell segmentation
To detect cells in an image and mark their contour, we used the Detectron2 deep neural network library (*Wu et al., 2019*), which relies on a Mask R-CNN image segmentation model (*He et al., 2018*) with the ResNet-101 (*He et al., 2015*) as its backbone.

### Model training and validation
Training the model required three rounds of manual annotation and training.

#### Initial manual annotation of the dataset and model training
We annotated cell contours manually using the VGG Image Annotator software (*Dutta and Zisserman, 2019*). Initially, we annotated only the hippocampus, which is relatively large and easily discernible. The hippocampus contains subregions of different densities, which we believed would adequately represent the variety of cell densities across the mouse brain. We manually annotated tiles of 312 × 312 pixels ($109 \times 109\mu m$), randomly selected from the hippocampus in five sections of three brains (55 tiles in total). We

**Table 1.** Breakdown of the data by strain and sex.

| Strain | Females | Males | Total |
|---|---|---|---|
| C57BL/6J | 174 | 195 | 369 |
| FVB.CD1(ICR) | 69 | 69 | 138 |
| Total | 243 | 264 | 507 |

provided these tiles to the network as training data, together with basic data augmentation (e.g., rotation and brightness changes) (*Krizhevsky et al., 2012*).

### Retraining on hippocampal sections
We then applied the trained model to detect cells on a new set of 55 randomly selected hippocampus tiles. We manually corrected the results produced by the network to create a new set of ground truth annotations. Next, we retrained the model from scratch over a combined training set of 110 tiles.

### Retraining on other regions
We subsequently used the trained model to detect cells on random sections of three selected brains. Visual inspection enabled us to select a set of 64 tiles that displayed the least accurate results and annotate them manually.

### Final training
We retrained the model from scratch on the resulting training set of 174 tiles (selected from ~15 sections of ~10 brains). The total number of cells across the training set tiles was 6247, corresponding to 0.008% of the estimated 77 million cells in the whole brain.

### Technical details
We conducted the training with a batch of size 2, a learning rate of 0.00025, with decay, using the Adam optimizer (*Kingma and Ba, 2014*). Training over 174 tiles required ~395,000 iterations and took ~36 hr using a Linux server with 160 Intel Xeon Gold 6248 2.5 GHz CPUs and a Tesla V100S-PCIE-32GB GPU.

### Evaluating model performance
The training process completed when the model converged. The accuracy of the model on the training data was 99.8%, with a false negative rate of 0.4%. To evaluate model performance, we manually annotated 30 additional tiles from the isocortex, medial amygdala (MEA), hypothalamus (HY), and hippocampus (HIP) of 27 brains and compared them with model prediction (*Table 2*). We obtained highly accurate results, comparable to the performance over the training data, for segmentation scores such as Jaccard measure (*Jaccard, 1912*), F1 score (harmonic mean of precision and recall), and total errors (i.e., percentage of mislabeled pixels), as well as for detection scores such as accuracy (detected cells divided by total cells) and false positive rate (false positives divided by total cells).

## Brain-wide automatic segmentation
The trained DNN was applied to 507 brains, as described in detail below.

## Extracting cell information per section
We divided each section into overlapping tiles sized 312 × 312 pixels, with an overlap of 20 pixels on each side (thus mitigating potential artifacts close to the borders of the tiles). We then applied the trained DNN to detect cells in each tile, resulting in a cell mask (i.e., a Boolean 312 × 312 matrix

**Table 2.** Model performance over out-of-sample tiles.

| Region | # cells in the test set | Segmentation (pixel-wise) scores | | | Detection (cell-wise) scores | |
|---|---|---|---|---|---|---|
| | | Jaccard index | F1 | Total errors | Accuracy | False positive rate |
| Isocortex | 192 | 0.982 | 0.991 | 0.002 | 0.962 | 0 |
| MEA | 115 | 0.975 | 0.987 | 0.001 | 0.962 | 0 |
| HY | 163 | 0.953 | 0.974 | 0.003 | 0.938 | 0.005 |
| HIP | 566 | 0.986 | 0.992 | 0.001 | 0.979 | 0.009 |

MEA: medial amygdala; HY: hypothalamus; HP: hippocampus.

whose entries are *true* if the corresponding pixel is part of a detected cell and *false* otherwise). Next, we stitched the tiles together using a logical OR over overlapping areas, resulting in a single cell mask per section. Subsequently, we performed contour detection to obtain the coordinates of each cell in a section and computed the morphological properties of each cell (i.e., circumference, diameter, and area). Following this analysis step, each section image was represented by a table containing the coordinates and morphological properties of its cells.

## Assigning cells to regions

We used the AMBA (*Wang et al., 2020*) to assign the coordinates of detected cells in each section to their corresponding brain region. But the atlas annotation was too coarse for several regions of interest, that is, CA1, CA2, and CA3 of the hippocampus. The common denominator of these regions was the presence of a dense and a sparse region that were not separated by the atlas (e.g., the pyramidal and stratum regions of CA1, CA2, and CA3). To provide the coordinates of these subregions, we defined a local measure of density referred to as cell 'coverage' and used it to cluster the relevant cells into a dense and a sparse region. Briefly, in a window of 64 ×64 pixels centered around each cell we counted the number of pixels that belong to cells, thus assigning a local 'coverage' measure (the median cell area was 80 pixels, much smaller than the window around it). We then detected the subregions by clustering the cells according to their 'coverage' values. For example, we took the 'coverage' values of all CA1 cells and used K-means clustering to split them into two clusters of high and low 'coverage' values. In this way, the coordinate of each cell center was assigned to either cluster. We then drew the circumference of the subregions by applying a standard morphological closing operation and discarded spurious small regions.

## Estimating volumes, 3D densities, and cell counts

Until this stage, the analysis provided local, that is, microscopic properties for each detected cell, and assigned cells to a brain region. The next step was to collect cells that belong to each region and estimate their density, the volume of the region, and the total cell count. This required calculating 3D estimates based on the relevant 2D data using the following steps:

### Estimating cell density per section

We used AMBA to label the area of a given region in a section. We assumed that cells belonging to a region are equi-radius spheres whose projection on the 2D section depends on the distance between their centers and the optical plane, and on the optical depth of field (*Figure 2—figure supplement 1*). Hence, detected cells on a 2D section $s$ originate from a slab whose volume is $v_s = a_s \cdot (2R + d)$, where $a_s$ is the area of a region, $R$ is the radius of the cells in the region, and $d$ is the optical depth of field. Cell density per section, $\rho_s$, is given by dividing the number of detected cells by $v_s$. The value of $a_s$ is measured by pixels whose size is $0.35 \mu m$, and $d = 1.5 \mu m$ (*Ragan et al., 2012*; *Amato et al., 2016*). The value of $R$ was taken as the 90th percentile of measured cell radii in $a_s$. The distribution of cell radii corresponds to the 'projection' of the cells on the measured section, together with the optical depth of field. Downstream results of cell count and density significantly depend of the value of $R$, for example, using the 50th percentile would provide larger estimated cell counts. Yet, rank order of cell counts and densities across regions is independent of the selected value of $R$.

### Calculating region volume

AMBA provides pixel-wise region annotation for each section, making it possible to calculate the area of a region per section (which is independent of cell segmentation). The 3D volume of a region is given by the sum of region volumes between adjacent sections, estimated by the average of its areas over each section. For example, if a region appears in sections 1, 2, 3, and 4, its volume is the sum of average volumes between sections 1 and 2, 2 and 3, and 3 and 4.

### Calculating cell counts across adjacent sections and in total

Cell counts between the adjacent sections of each region are given by the average densities in those slides multiplied by the volume of the region between these sections. The total cell count of a region is provided by a sum across all relevant sections.

## Calculating cell densities per region

The overall density of each region is given by the total cell count divided by the volume of the region.

## Discarding whole brains or particular regions of lower technical quality

After calculating the 3D counts and densities across all regions in all brains, we excluded from subsequent analysis regions and whole brains that displayed potentially flawed estimates. We applied the following criteria:

### We discarded brains displaying dark images

We filtered out brains whose median brightness across the whole brain gray matter ('gray' region) was lower than 25 (on a scale between 0 and 255). In such cases, all ~140 sections of the brain were excluded from downstream analysis because DNN cell detection was either impossible or provided significantly lower estimates (*Figure 1—figure supplement 2A*). In total, 51 brains were discarded due to low brightness.

### We discarded brains displaying an optical or physical artifact resulting in outliers in cell count

We noticed that a common optical or physical artifact of resolution degradation caused the DNN to falsely detect large amounts of excess cells (see distinct 'columns' in *Figure 1—figure supplement 2B*). We marked cases in which cell count in a region was 3 SDs larger than the median for the region across brains (calculated as $MAD \cdot 1.4826$, assuming normal distribution). We discarded brains that included more than three such outlier regions. 81 brains were discarded due to such artifacts.

### We discarded regions of small volume

We filtered out regions whose median volume across brains was smaller than $0.3 mm^3$, or whose median cell count across sections was smaller than 500 (*Figure 1—figure supplement 2C*). In total, 283 (out of 690 regions in AMBA) were discarded and excluded from all brains (data of these regions do appear in our online tool described below).

### We discarded regions displaying different estimates in right vs. left hemispheres

Cell count estimates in the right and left hemispheres served as a proxy for technical noise. We calculated cell counts per region using each hemisphere independently. If the average difference in cell count between hemispheres was higher than 15.5% of the total cell count for that region, we excluded the case from downstream analysis (*Figure 1—figure supplement 2D*). 33 regions were omitted due to right–left differences.

### We discarded regions displaying a correlation between cell count and image brightness

We excluded regions exhibiting strong correlation (>0.3) between brightness and cell count because we assumed that in this case cell count was affected by the inability of the model to discern the cells when the brightness was too low. 30 regions were discarded due to such correlations.

In sum, we processed 507 brains, of which 132 were fully discarded. Of 690 regions in AMBA, 346 were discarded completely and another 107 were partially discarded.

## Correcting a batch effect in volume

While analyzing the Allen images, we noticed that the experiments were performed in several batches (based on experiment ID), and that there was a small difference in the total brain volume among batches, yet the total cell did not display a batch effect (*Figure 2—figure supplement 1B and C*). In order to correct this effect, we multiplied each brain volume by a batch-specific factor such that the batch median volume 8 will be 380 mm³. These factors were 1.064, 1.032, and 1.008 for batches 1, 2, and 3, respectively. This correction subsequently effected the density and cell area estimates but did change cell counts.

## Predicting sex based on brain region volume or density

### Performance estimates based on a linear SVM

We randomly selected 2/3 of the C57BL/6J brains (n = 184), where each brain is represented by a 532-dimensional vector of either volume or density across regions, and trained the SVM (C = 100). We then tested its performance on the left out 1/3 of brains (n = 97). These random splitting into a training and test sets was repeated 100 times. The performance was 0.910 ± 0.022 and 0.780 ± 0.039 for volume and density, respectively.

### SVM based on specific regions

Each of the regions was ranked according to the average absolute value of its weight in the separating hyperplane across the 100 SVMs for either volume or density. The top 10 regions (i.e., SVM features) were selected and an SVM was trained on the whole dataset (i.e., without a train-test split) while adding one region at a time.

## Mice

Adult, C57BL/6JOlaHsd wildtype mice (Envigo) were housed under standard conditions and provided chow and water ad libitum. Experimental procedures followed the legislation under the Israel Ministry of Health – Animal Experiments Council and were approved by the institutional Animal Experiments Ethics Committees at Technion Israel Institute of Technology (Permit #IL1190819).

## Histology

One male mouse was sacrificed by an overdose of ketamine/xylazine, followed by transcardial perfusion with phosphate-buffered saline (PBS), then 4% PFA in 1× PBS. The brain was extracted, postfixed 24 hr in 4% PFA at 4°C, cryoprotected in 15%, then 30% sucrose, and finally OCT-embedded, frozen and cryosectioned (20 µm). Sections were collected free-floating to PBS. For staining, we first incubated sections 1 hr in blocking buffer (5% normal goat serum, 3% BSA, 0.3% Triton X-100, in PBS with 0.05% sodium azide), followed by overnight incubation in 1:500 anti-NeuN-PE (Milli-Mark A60, #FCMAB317PE), in the same blocking buffer. After three 10 min washes in PBS, we counterstained 3 min with DAPI (NucBlue, Thermo #R37606), mounted, and coverslipped sections (Fluoromount-G, Invitrogen #00-4958-02), and imaged on a Nikon Ti2 Eclipse inverted fluorescence microscope (×20 objective).

## STPT on nuclei-stained brains

### Perfusion

Nine adult female C57BL/6JOlaHsd mice were euthanized via intraperitoneal injection with an overdose of ketamine/xylazine mixture (100 and 10 mg/kg, respectively). The mice were transcardially perfused with PBS followed by freshly prepared PFA 4%. Subsequently, brains were removed and postfixed in PFA 4% solution at 4°C for 24 hr and then transferred to PBS containing 0.05% sodium azide at 4°C until used.

### Nuclear staining

The brains were Hoechst stained according to *Seiriki et al., 2017*. Briefly, the brains were immersed in staining solution (PBS with 0.05% sodium azide containing 10 ug/ml Hoechst 33342 Trihydrochloride, Trihydrate [Invitrogen #H3570], with 0.1% Triton X-100, and 0–30% w/v sucrose) for 8 d (sucrose concentrations: day 1–10%, day 2–20%, days 3–7–30%, and day 8–0%) at 55°C with gentle shaking and then returned to 4% PFA in PBS for 24 hr, then kept in PBS with 0.05% sodium azide until used.

Brains were sent to TissueVision Inc (Boston) for imaging on a TissueCyte 1000 system with ×16 objective (estimated resolution of 0.7 µm per pixel), with a gap of 100 µm between coronal sections.

## Statistical tests

All statistical tests were based on the nonparametric Wilcoxon rank-sum test, thus not assuming a certain underlying distribution. Resulting p-values were then corrected for multiple testing by the Benjamini–Hochberg procedure.

## Materials availability statement

The images and the corresponding atlas registration data were retrieved from the Allen Connectivity Project portal using Allen SDK for Python at https://allensdk.readthedocs.io/en/latest/.

Segmentation-related code is available at https://github.com/delkind/detectron_segmentation (copy archived at *Elkind, 2023a*).

## Brain Explorer: Data viewer for mouse brain cell data

We provide a tool for examining analyzed AMBCA brain images. The tool allows visualizing distributions of macroscopic parameters (i.e., density, volume, cell count, etc.) for any region across any subset of the brains, as well as performing statistical tests. Subsets of brains can be based on any combination of strain and sex. Resulting charts can be exported as CSV or PDF files. The viewer is available in https://github.com/delkind/mouse-brain-cell-counting (copy archived at *Elkind, 2023b*).

## Acknowledgements

NS is funded by the Ministry of Science, Technology & Space, Israel (grant 3-16033). AZ is supported by the European Research Council (TYPEWIRE-852786), Human Frontiers Science Program (CDA-0039/2019C), and Israel Science Foundation (2028912). HH is supported by the Swedish Brain Foundation (Hjärnfonden) and Human Frontiers Science Program (CDA-0039/2019C). We thank Eytan Domany for critical discussion. We thank Dvir Aran and the Open University for computing resources.

## Additional information

### Funding

| Funder | Grant reference number | Author |
|---|---|---|
| European Research Council | TYPEWIRE-852786 | Hannah Hochgerner Etay Aloni Amit Zeisel |
| Human Frontier Science Program | CDA-0039/2019-C | Hannah Hochgerner Amit Zeisel |
| Israel Science Foundation | 2028912 | Hannah Hochgerner Amit Zeisel |
| Swedish Brain Foundation | | Hannah Hochgerner |
| Israel ministry of science, technology & space | 3-16033 | Noam Shental |

The funders had no role in study design, data collection and interpretation, or the decision to submit the work for publication.

### Author contributions

David Elkind, Data curation, Formal analysis, Investigation, Methodology, Writing - original draft; Hannah Hochgerner, Investigation, Visualization, Writing - original draft, Writing - review and editing; Etay Aloni, Investigation, Methodology; Noam Shental, Formal analysis, Supervision, Funding acquisition, Investigation, Methodology; Amit Zeisel, Conceptualization, Formal analysis, Supervision, Funding acquisition, Methodology, Writing - original draft

### Author ORCIDs

Noam Shental (ID) http://orcid.org/0000-0001-9645-3773
Amit Zeisel (ID) http://orcid.org/0000-0002-2424-9279

### Decision letter and Author response

Decision letter https://doi.org/10.7554/eLife.82376.sa1
Author response https://doi.org/10.7554/eLife.82376.sa2

## Additional files

### Supplementary files
- MDAR checklist

### Data availability
All data generated or analysed during this study are included in the manuscript and supporting file; Tables related to values of data appear in the figures can be found in excel file.

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
