## [Editor Report]

The manuscript provides a new powerful tool as well as a large resource that should be useful both to the neuroscience community and more widely. The authors developed and applied a methodology to automatically estimate volume, cell number, and density of mice brains from multiple regions, by detecting the auto-fluorescence intensities of the cell nuclei. Using this platform, they analyzed a few hundred mouse brains available in the database of the Allen Mouse Brain Connectivity project. They identified strain-specific and sex-specific differences in several brain regions.

---

## [Decision Letter]

**Decision letter after peer review:**

Thank you for submitting your article "Sex, strain and lateral differences in brain cytoarchitecture across a large mouse population" for consideration by *eLife*. Your article has been reviewed by 3 peer reviewers, and the evaluation has been overseen by a Reviewing Editor and Catherine Dulac as the Senior Editor. The reviewers have opted to remain anonymous.

In its current form, the manuscript seems to better fit a "Tools and Resources" article. This would also require an essential revision, including some validation data, corrections, and clarification of technical issues, as well as better descriptions of the previous relevant literature and improved explanations of the data analysis.

In your revision, as you address the Reviewers' points, please consider these essential revisions:

1) Please clarify what and why some data is discarded from the results.

2) Add some validation data of the main findings by using classical neuroanatomical staining (H&A or immunostaining) of independent mouse brains to quantify volume, cell number, and density in brain regions where sex differences or strain differences were identified.

3) It would be helpful to include figures that summarize the overall main findings.

4) Please include and discuss previous literature, which reports the native fluorescence described by Webb's group.

5) I suggest submitting the revised article as a "tools and resources" article.

*Reviewer #1 (Recommendations for the authors):*

My concerns largely focus on:

1) The native fluorescence and second harmonic generation from multiphoton illumination can accurately be used to quantify cell nuclei because this is not what the signal is generated from.

2) The authors should validate their findings in an independent dataset generated with adequate labeling methods (e.g. DAPI, NeuN, etc.).

*Reviewer #2 (Recommendations for the authors):*

The weakness of this manuscript is the overall message. While I really appreciate their effort and conclusions to resolve some of the fundamental questions in brain science, such as are cell numbers different between males vs females, these data seem rather descriptive rather than mechanistic. For example, authors could do some hypothesis testing such as masculinizing females or using pregnant female brains to study changes in brain size. As this manuscript stands, this type of data might find a better outlet in a more brain anatomy-focused journal.

Specific points:

1. I find the figures can be improved.

- Some of the plots have legends that are illegible (e.g., Figure 1 F, Figure 2 A,B,F, Figure 3 B,C, Figure 4B,C,D,E).

2. Figure 3: it looks like there are many areas that are sexually dimorphic in either volume or/and density, but in the main text only a handful of areas were featured. Is there a specific trend as to which areas tend to be subject to sex-specific regulation? For example, can authors roughly align this dataset with Allen Brain Atlas to find correlations between the expression of gonadal steroid hormone receptors and the cell numbers?

*Reviewer #3 (Recommendations for the authors):*

1. In Figure S1B (esp. regions III and IV) there seems to be a number of false positive detections, i.e. detections which do not overlap with DAPI. The authors state say that they find an 'overlap of nearly 100% with nuclear staining (DAPI)', but they should quantify specificity and selectivity based on their existing data here.

2. Figure 1E: What is the number of samples here? It would be better to plot these values as mean +/- SEM.

3. Figure 1B, C: scale bars missing.

4. In Figure 1D: the acronyms should be defined in the legend.

5. Figure 3A and Figure 5B could be removed in my opinion since it does not add any relevant information.

6. Figure 3D: it is unclear why these particular areas were selected.

7. Figure 4 is very hard to understand and extremely dense.

8. Figure 5C: why are AAA and SSs2/3 (highlighted in D, E) not shown in this plot?

9. The two selected areas in Figure 5D, and E are not present in Figure 5C.

10. Several figure panels are overplotted to such a degree that it is impossible to decipher brain area acronyms (e.g. Figure 2A, 3C). It might be helpful to highlight the relevant ones mentioned in the text. In Figure 3C, many markers of statistical significance are very hard to distinguish.

11. On several figure axes, characters do not seem to render (e.g. µ): Figure 1C, Figure 3C, Figure S2C, etc.

12. Figure S1A would benefit from a quantitative legend for the heatmap and S1B from a y-axis.

13. Figure S2: This figure is confusing since the brain sections on the left do not match the ones displayed on the right. Also, panel A needs a quantitative heatmap. In panels B and C, y axes. are missing. In panels B and C, layer 4 (L4) is missing from the first three cortex distributions.

14. The Allen Brain Atlas only contains part of the olfactory bulb and part of the cerebellum. As a result, (1) registration accuracy might be lower in these regions, and (2) the cell counts reported here are likely to underestimate the actual numbers. I think this should be more prominently discussed.

15. For cell numbers, the authors find a striking standard deviation between individual males of ~10 million cells. It would be helpful to report this number for females, too.

16. An open question is – of course – what fraction of cells are in fact neurons. Have the authors explored whether any features of the detected low autofluorescence objects have distributions that match the distributions of neurons vs glia in areas where their rel. numbers are known?

17. The authors find that for 'some regions, cell count does not scale simply or linearly with volume' (line 482). Have they explored non-linear models to fit these data?

18. In line 149, it should be made clearer what 'levels 6-8 of the region hierarchy' refers to.

19. The abstract states: 'We applied our pipeline to 537 brains […]' but this is slightly misleading since 399 brains were actually part of the analysis according to the methods section.

20. Line 290: what does "grey" refer to here?

21. Line 311: the statement 'where most brain regions fell in quadrant IV of the volume-density plane' is somewhat cryptic.

22. Line 394: missing word? Grey matter.

23. Line 589: references should be in ascending order here.

---

## [Author Response]

In your revision, as you address the Reviewers' points, please consider these essential revisions:1) Please clarify what and why some data is discarded from the results.

We either discarded whole brains or specific regions (across all brains). In summary, 26% of the brains were discarded, mostly due to poor imaging conditions that would have skewed our quantifications. In addition, 50% of the regions were discarded from our analysis due, e.g., to their small volume (we note that these regions still appear in our online tool). We elaborate on this issue in our response to a specific question by Reviewer #1, and we have accordingly revised the manuscript’s Methods section: “Discarding whole brains or particular regions of lower technical quality”.

2) Add some validation data of the main findings by using classical neuroanatomical staining (H&A or immunostaining) of independent mouse brains to quantify volume, cell number, and density in brain regions where sex differences or strain differences were identified.

In our initial Supplemental Figure 1 (in current version Figure 1—figure supplement 1) we provide some technical validation of the method, by showing nuclear staining, and autofluorescence side-by-side, using epifluorescence microscopy. As rightfully pointed out by Reviewer 3, in our revision, we now report appropriate statistical measures for this analysis (true positives, false positives, false negatives).

In addition, we performed the following two sets of validations –

(i) Technical validation of our staining-free quantification approach, by nuclear staining. We performed nuclear staining (Hoechst 33342) followed by STPT imaging of 9 female brains and trained a new deep neural network (DNN) to segment the resulting images (STPT was performed by TissueVision, using similar technology applied by AMBCA). Unfortunately, in STPT it is not technically possible to analyze nuclear staining and autofluorescence in the very same tissue. Therefore, we compared per-region density, cell count and volume of the nuclei-stained validation brains to our original DNN-based analysis of AMBCA brains. We show a correlation coefficient >0.99 for per-region cell count in AMBCA autofluorescence and our nuclear staining (and a similar correlation coefficient for volume). However, the number of cells in nuclear staining over the whole brain is 56% larger than in autofluorescence. Although we have no technically feasible way to prove this, one likely explanation for this discrepancy is the nature of the detected signals; as “positive” (Hoechst fluorophore) or autofluorescence. Further, discrepancies between the two methods were notably higher in glial-rich tissues (e.g., CTX L1, midbrain, brainstem) – thus one may speculate that low-autofluorescent object counts may be biased to detect neurons, rather than glia.

(ii) Independent validation of the biological findings. To validate region-specific sex differences in volume, we processed an additional independent cohort of 829 C57BL/6J brains (663 males and 166 females) from the Allen Mouse Brain Connectivity Project. We show, e.g., that sexual dimorphism in MEA, BST and ORBvl2/3 is recapitulated (see Results section “Population-wide, regionally resolved exploration of neuroanatomical features” and Figure 4—figure supplement 1).

3) It would be helpful to include figures that summarize the overall main findings.

We have included a graphical abstract as new Figure 1.

4) Please include and discuss previous literature, which reports the native fluorescence described by Webb's group.

We now specifically refer to Webb’s work in the first section of the Results “Autofluorescence of STPT images displays cell nuclei” and the Discussion, as described in our response to Reviewer #1.

5) I suggest submitting the revised article as a "tools and resources" article.

We agree and are happy to submit the revised version as a “tools and resource” article.

We would like to also mention that during the revision we detected a batch effect in the AMBCA experiments, affecting the volume in their first batch of experiments. This batch which included mostly males had, for some reason, lower overall volume. We emphasize that (1) the total number of cells did not show any batch effect; (2) We normalized the volume and repeated the analysis. Results remained mostly unchanged, except for our previous finding regarding the overall larger volume of BL6 females.

In addition, the Reviewers mentioned graphics issues in many of the figures. We agree with these comments. Many panels were overplotted to a degree that region acronyms were hard to decipher. We went over all panels, deleted region acronyms when overlaying other region names, added arrows or lines pointing to regions that are mentioned in the text, etc. We therefore hope that all panels are now clearer. For completeness, we provide a supplementary table that include data of relevant panels. We hope this will bridge most of the graphical issues.

Reviewer #1 (Recommendations for the authors):My concerns largely focus on:1) The native fluorescence and second harmonic generation from multiphoton illumination can accurately be used to quantify cell nuclei because this is not what the signal is generated from.

We hope our reply in the Public Review section, and our validation experiments included in the revised manuscript sufficiently addressed the Reviewer’s concern on the technical validity of our approach. We hope we also clarified any doubts on correspondence of our data to Murakami’s results, by reporting the statistical measures the Reviewer asked for.

2) The authors should validate their findings in an independent dataset generated with adequate labeling methods (e.g. DAPI, NeuN, etc.).

We performed technical validation of our results – with full brain SPTP imaging of 9 Hoechst-stained brains, and compared the results to AMBCA autofluorescence. We added a relevant passage, and Figure 2—figure supplement 2; see below. As expected, autofluorescence-based cell counting performed better in the e.g., isocortex, compared to more dark regions in the hindbrain.

Text added to Results section “Population-wide, regionally-resolved exploration of neuroanatomical features”:

“To expand on our technical comparison between quantification based on autofluorescence vs. staining (Figure 1—figure supplement 1), we performed nuclear staining (Hoechst 33342) and whole-brain STPT imaging on nine brains of female C57BL/6J mice. We trained a DNN of the same topology using tiles from the resulting Hoechst images, registered with the Allen Mouse Brain Atlas coordinates, and repeated our per-region estimates of volume, density, and cell count (see Methods). As expected, correlation between this in-house dataset, and the AMBCA analysis was very high when comparing the volume across regions (r = 0.99, Figure 2—figure supplement 2A). In addition, cell count correlations were also high (r = 0.99, Figure 2—figure supplement 2B), but median cell count in Hoechst was 65% higher; recapitulating the trend toward false negatives observed in Figure 1—figure supplement 1 (epifluorescence microscopy). However, agreement in cell density varied by region. The correlations in density were fair across cortical regions of layers 2/3, 4, 5 and 6a (r = 0.78, Figure 2—figure supplement 2C1): they displayed comparable densities between the methods (Figure 2—figure supplement 2D1). The correlation value across cortical regions 1 and 6b was similar (r = 0.79), yet the error in absolute values was significantly higher (Figure 2—figure supplement 2C2), at about 2-fold higher density using Hoechst (Figure 2—figure supplement 2D1). The correlation across brain stem regions was also lower (r = 0.56, Figure 2—figure supplement 2C3), yet rank order across regions was similar to cortical regions 1 and 6b. Cerebellar regions, which are significantly denser, displayed a 2-fold density difference (r = 0.65, Figure 2—figure supplement 2C4). This detection bias may therefore stem from several underlying causes; (1) inaccurate registration of border regions (CTX L1 and 6b), (2) physical detection limits of the low-autofluorescent objects compared to stained objects, and (3) region-, or cell type-inherent differences in autofluorescence, resulting in lower detection of cells in glial-rich (e.g., CTX L1, brainstem), compared to neuron-rich (e.g., other cortex layers) regions.”

We entirely agree that this validation was necessary for confidence in our findings. We also hope the Reviewer appreciates the time and financial efforts that were required to perform this validation, including the establishment of a whole-brain Hoechst staining protocol, STPT imaging abroad, and training an entirely new DNN for the analysis of this dataset.

Since the analyses we present mostly focus on comparisons across brains using a large, unique, consistent cohort, systematic errors have minor effects on the main findings.

Therefore, in light of the scope of our study, we hope the Reviewer also finds the combination of our technical (experimental) validation, and validation of our central findings on an independent AMBCA cohort – including power analysis – both sufficient to our conclusions, and inviting for further exploration in the community.

Reviewer #2 (Recommendations for the authors):The weakness of this manuscript is the overall message. While I really appreciate their effort and conclusions to resolve some of the fundamental questions in brain science, such as are cell numbers different between males vs females, these data seem rather descriptive rather than mechanistic. For example, authors could do some hypothesis testing such as masculinizing females or using pregnant female brains to study changes in brain size. As this manuscript stands, this type of data might find a better outlet in a more brain anatomy-focused journal.

We agree that our study is descriptive rather than mechanistic. We think, however, that it provides a systematic quantitative approach over hundreds of brains, yielding biologically relevant findings. Taking these findings to a functional level is beyond the scope of the current study – and we do hope to encourage further interesting exploration along the lines suggested by the Reviewer.

Specific points:1. I find the figures can be improved.– Some of the plots have legends that are illegible (e.g., Figure 1 F, Figure 2 A,B,F, Figure 3 B,C, Figure 4B,C,D,E).

We agree. We hope the changes to our revised manuscript, such as omissions of datapoint labels from figures, have improved legibility. For complete presentation of this large dataset, we instead provide all results as Supplemental tables.

2. Figure 3: it looks like there are many areas that are sexually dimorphic in either volume or/and density, but in the main text only a handful of areas were featured. Is there a specific trend as to which areas tend to be subject to sex-specific regulation? For example, can authors roughly align this dataset with Allen Brain Atlas to find correlations between the expression of gonadal steroid hormone receptors and the cell numbers?

We thank the Reviewer for this interesting suggestion. We found a tendency of amygdala-related regions to have larger volume or be denser in males, while regions of the prefrontal cortex (mainly in the ORB) were found to be larger and denser in females. We have added the following sentence to the Abstract.

We also checked whether gonadal receptor genes have higher expression in regions found to be sexually dimorphic. We used a voxel-based expression dataset that is based on quantifications of the Allen mouse brain ISH atlas. To the best of our knowledge these data are based on P56 males only (although some ISH are available for females their quality was not sufficient).

We used the Allen Mouse Brain ISH Atlas API, that contains P56 males only (http://mouse.brain-map.org), and tested the following genes : *Ghr, Ghrhr, Pgr, Fshr, Ar, Esr1, Esr2, Lhcgr, Ghsr, Gnrhr, Thra, Thrb, Lepr* and *Prlr*. Unfortunately, only five of these genes had sufficiently high ISH data quality. We calculated their average expression per region and present it in the heatmap below. For example, prolactin receptor (*Prlr*) is highly expressed in the hypothalamus and amygdala. Interestingly, Thyroid Hormone Receptor subunits (*Trha, Trhb*) have higher expression in frontal cortical areas. While this can be considered an interesting anecdote, we do not feel that this analysis is sufficiently deep or conclusive to claim that gonadal hormone receptors drive sexual dimorphism in volume or cell density, and did not include this analysis in the manuscript.

**Author response image 1. sa2fig1:** Heatmap of ISH data for five genes across brain regions .

Reviewer #3 (Recommendations for the authors):1. In Figure S1B (esp. regions III and IV) there seems to be a number of false positive detections, i.e. detections which do not overlap with DAPI. The authors state say that they find an 'overlap of nearly 100% with nuclear staining (DAPI)', but they should quantify specificity and selectivity based on their existing data here.

We thank the Reviewer for this important comment. The current text mentions true positives, false positives and false negatives.

Updated section:

“To confirm that these dark objects indeed represent cell nuclei with lower autofluorescence intensity than the surrounding lipid-rich brain tissue, we performed a standard 4% PFA perfusion-fixation followed by cryosectioning and nucleus (DAPI) counterstaining. In these sections, we observed the same low-autofluorescent objects using epifluorescence microscopy. 91%-98% of detected objects were also marked by DAPI, confirming that dark objects in STPT indeed represent cell nuclei (Figure 1—figure supplement 1). About 2%-9% of detected low-autofluorescent objects were additional “false positive” detections that were not obvious in the DAPI image. On the other hand, 26%-45% of DAPI-detected nuclei were not observed in the autofluorescent images (false negatives), pointing to an underestimate of cell counts based on low-autofluorescence objects, compared to nuclear staining. A more systematic comparison between autofluorescence images and nuclear staining appears in the next section.”

2. Figure 1E: What is the number of samples here? It would be better to plot these values as mean +/- SEM.

We added the number of samples (195). We added SEM and now mention it in the legend. However, their values are very small and thus appear as thin red bars.

3. Figure 1B, C: scale bars missing.

Thank you, scale bars were added in Figure 1B, Figure 2A.

4. In Figure 1D: the acronyms should be defined in the legend.

True, this was corrected.

5. Figure 3A and Figure 5B could be removed in my opinion since it does not add any relevant information.

We have removed Figure 3A. We agree with the Reviewer that the examples in “Figure 5B” (now Figure 6B) do not add new information, yet we prefer to keep them for the sake of providing an example.

6. Figure 3D: it is unclear why these particular areas were selected.

We selected examples of regions where males’ volume was either larger or smaller than in females, while density was either the same (BST) or also different (ORBvl2/3). We also show an example of insignificant volume differences yet a significant difference in density is observed (AUDv2/3). We now justified their presentation in the text.

7. Figure 4 is very hard to understand and extremely dense.

We agree that Figure 4 (now Figure 5) is hard to understand both because it contains several comparisons (across strains and across sex; showing volume and density), and also due to graphics issues. We have tried to ease the graphical burden by (a) increasing the size of problematic panels; and (b) deleting region acronyms when unreadable.

We also provide a Source data file containing all underlying data. We understand that this is far from ideal, but we believe the combination of a “clearer” figure that allows zooming in, and a file that contains the same data would provide readers adequate means of exploring the data.

8. Figure 5C: why are AAA and SSs2/3 (highlighted in D, E) not shown in this plot?

Thank you, this is now corrected.

9. The two selected areas in Figure 5D, and E are not present in Figure 5C.

Thank you, this is now corrected.

10. Several figure panels are overplotted to such a degree that it is impossible to decipher brain area acronyms (e.g. Figure 2A, 3C). It might be helpful to highlight the relevant ones mentioned in the text. In Figure 3C, many markers of statistical significance are very hard to distinguish.

We agree, and we hope our changes have improved readability.

11. On several figure axes, characters do not seem to render (e.g. µ): Figure 1C, Figure 3C, Figure S2C, etc.

Thank you, this is now corrected.

12. Figure S1A would benefit from a quantitative legend for the heatmap and S1B from a y-axis.

We believe the Reviewer referred to Figure S2 (in current version Figure 2—figure supplement 3). Regarding panel A, we added a quantitative legend. Panel B displays a (probability) distribution and therefore y-axis is omitted.

13. Figure S2: This figure is confusing since the brain sections on the left do not match the ones displayed on the right. Also, panel A needs a quantitative heatmap. In panels B and C, y axes. are missing. In panels B and C, layer 4 (L4) is missing from the first three cortex distributions.

We agree with the Reviewer that brain sections in panel A do not match those in panels B and C. We did not mean to create a linkage between panels A and B+C. We have moved panel A above panel B and C, which we believe creates the appropriate separation and breaks the unwanted linkage. Regarding panel A, we added a quantitative legend. Panel B-C displays a (probability) distribution and therefore y-axis is omitted. As for L4, while visual and somatosensory cortices contain L4, in motor and orbital cortices L4 is lacking.

14. The Allen Brain Atlas only contains part of the olfactory bulb and part of the cerebellum. As a result, (1) registration accuracy might be lower in these regions, and (2) the cell counts reported here are likely to underestimate the actual numbers. I think this should be more prominently discussed.

Thank you for this. We agree that the STPT image data analyzed do not cover the whole olfactory bulb and cerebellum, and their cell counts may be lower. We therefore explored whether olfactory bulb and cerebellum data is consistent across brains, and can be fairly included in our systematic analysis. Indeed, MOB and CB data were slightly “noisier” than other regions, as demonstrated by how variance in volume in these regions is somewhat higher compared to more interior regions with similar volumes (see figure below; these panels were also added to Figure 2—figure supplement 1D-E). We now include the following clarification in the Results section “Population-wide, regionally-resolved exploration of neuroanatomical features”, and added a statement to the Discussion:

Results:

“The anterior-most olfactory bulb (MOB) and posterior-most cerebellum (CB) were truncated in imaging, which likely led to an underestimate in their quantifications, and slightly higher variance compared to other regions of similar volume (Figure 2—figure supplement 1D-E). “

Discussion:

“The study has several limitations. (…) Third, regions at the anterior-posterior edges, i.e., olfactory bulb (MOB) and cerebellum (CB), appear partially truncated, and we likely report an underestimate of their parameters. MOB and CB volumes also displayed slightly higher variance than more interior regions with similar volume. Therefore, their population comparisons would appear noisier (although likely not biased), compared to other regions.”

15. For cell numbers, the authors find a striking standard deviation between individual males of ~10 million cells. It would be helpful to report this number for females, too.

Cell count (in millions) for males is 75±10, and for females is 77±9. We added the STD to Figure 4A for both males and females. Source data file table details the median and STD for volume, density and count per region for males and female, for both C57BL/6J and FVB.CD1.

16. An open question is – of course – what fraction of cells are in fact neurons. Have the authors explored whether any features of the detected low autofluorescence objects have distributions that match the distributions of neurons vs glia in areas where their rel. numbers are known?

We fully agree, and thank the Reviewer for this question, yet we cannot currently substantiate any of our speculations, as we outlined in the Public Review. The question of the type of cells that are “enriched” in STPT remains to be answered.

17. The authors find that for 'some regions, cell count does not scale simply or linearly with volume' (line 482). Have they explored non-linear models to fit these data?

We have not explored fitting the data by non-linear models. Our statement was “softer” and more “qualitative”, trying to state that regions are not scaled in a uniform manner. Hence, it might be that regions scale linearly, yet each has its own scale factor. We therefore replaced the word “linearly” by “uniformly”.

Former sentence (line 482): “This suggests that for some regions, cell count does not scale simply or linearly with volume.”

Current sentence: “This suggests that across regions cell count does not scale uniformly with volume.”

18. In line 149, it should be made clearer what 'levels 6-8 of the region hierarchy' refers to.

Levels 6-8 were taken from the AMBA convention for region hierarchy. We changed the sentence accordingly. For example, the AMBA hierarchy level 1 corresponds to the whole grey matter; Isocortex is level 5; and individual cortical layers are level 8.

Former sentence (line 149):

“In total, we estimated per-region D and V for 532 basic regions annotated in the AMBA, which corresponds to levels 6-8 of the region hierarchy.”

Current sentence:

“In total, we estimated per-region D and V for 532 basic regions annotated in the AMBA, which corresponds to levels 6-8 of the AMBA region hierarchy.”

19. The abstract states: 'We applied our pipeline to 537 brains […]' but this is slightly misleading since 399 brains were actually part of the analysis according to the methods section.

We apologize for being unclear. The DNN was applied to 537 brains, yet only 507 of them were C57BL/6J and FVB.CD1. We then excluded 132 brains from our analysis, as explained in the Methods section (see also answer to Reviewer #1).

20. Line 290: what does "grey" refer to here?

Corrected to: “whole brain grey matter”.

21. Line 311: the statement 'where most brain regions fell in quadrant IV of the volume-density plane' is somewhat cryptic.

We deleted the panel to which the sentence was referring.

22. Line 394: missing word? Grey matter.

Thank you, this was corrected.

Former sentence (line 394): “Moreover, within each strain, body weight did not correlate with grey volume.”

Current sentence: “Within each strain, body weight did not correlate with grey matter volume.”

23. Line 589: references should be in ascending order here.

Corrected.